

# Racialized bias in pediatric pain: the role of observers' attentional processing and estimations of children's pain

Ama Kissi[1], Dimitri Van Ryckeghem[1,2,3], Peter Mende-Siedlecki[4], Adam Hirsh[5], Ischa Van Alboom[1], Dries Debeer[6] and Tine Vervoort[1]

[1] Department of Experimental-Clinical and Health Psychology, Ghent University, Ghent, Flanders, Belgium
[2] Department of Clinical Psychological Science, Maastricht University, Maastricht, Netherlands
[3] Department of Behavioral and Cognitive Sciences, University of Luxembourg, Esch-sur-Alzette, Luxembourg
[4] Department of Psychological & Brain Sciences, University of Delaware, Newark, United States
[5] Department of Psychology, Indiana University-Purdue University at Indianapolis, West Lafayette, United States
[6] Department of Experimental Psychology, Ghent University, Ghent, Belgium

## ABSTRACT

**Background:** Research demonstrates racism in pediatric pain care. However, the mechanisms underlying these injustices are not well understood. This study examined White observers' attentional processing of facial expressions of pain demonstrated by White *vs.* Black children and observers' estimations of the pain expressiveness levels of these children. Furthermore, we assessed whether differences in observers' attentional processing were influenced by observers' pain beliefs and the pain expressiveness level.

**Method:** Eighty White adults (42 women; 38 men) performed the visual search task (VST), rated the levels of pain that the children expressed, and reported their beliefs concerning the pain experience of White *vs.* Black children.

**Results:** Findings revealed facilitated attentional engagement towards Black compared to White child pain faces, particularly at high pain expressiveness levels. No attentional disengagement effects were observed. Pain estimations increased with increasing pain expressiveness but, contrary to prior findings, did not differ for White *vs.* Black children. Observers' false pain beliefs did not significantly impact their attentional processing nor pain estimations.

**Conclusions:** The results underscore the importance of understanding how racialized disparities in observers' attentional processing of others' pain may contribute to racialized inequities in pediatric pain care.

Corresponding author
Tine Vervoort,
Tine.vervoort@ugent.be

[1] In line with *Morais et al.*'s *(2022)* recommendation, we use the term racialized rather than racial to acknowledge the socio-political process of racialization through which people in power assign arbitrary privilege to groups of people that have characteristics that are deemed valuable while other groups of people are marginalized and viewed as inferior for not possessing these characteristics.

# INTRODUCTION

Accumulating evidence documents racialized[1] inequities in pain care and the negative effects of unfair pain treatments on health (*Anderson, Green & Payne, 2009*; *Meghani, Byun & Gallagher, 2012*; *Meints et al., 2019*). For instance, research revealed that Black adults are more likely to have their pain underdiagnosed, receive fewer and weaker pain treatments, and have shorter medical encounters compared to their White counterparts (*Anderson, Green & Payne, 2009*; *Green et al., 2003*; *Hirsh et al., 2019*; *Meghani, Byun & Gallagher, 2012*; *Meints et al., 2019*; *Morales & Yong, 2021*; *Shavers, Bakos & Sheppard, 2010*). Such inequities have been observed in both acute and chronic pain care (*Anderson, Green & Payne, 2009*) and are viewed as fundamental drivers of health disparities (*Phelan & Link, 2015*). A growing body of research suggests that racism also exists in pediatric pain (*Goyal et al., 2015*; *Groenewald et al., 2018*; *Guedj et al., 2021*; *Johnson et al., 2013*). For example, *Guedj et al. (2021)* found that, compared to White children, Black and Hispanic children are less likely to receive opioid analgesia for their pain. Likewise, research by *Goyal et al. (2015)* revealed that Black children are less likely to achieve analgesia administration compared to White children. These inequities are consequential, as poorly managed pain in childhood predicts pain problems later in life (*Brattberg, 2004*).

Although racism in pain care has been well-documented across different ages and settings (*Anderson, Green & Payne, 2009*; *Hampton, Cavalier & Langford, 2015*; *Morales & Yong, 2021*; *Overstreet et al., 2022*), and theoretical accounts of potential explanatory mechanisms have been put forward (*e.g.*, *Mathur et al., 2022*), research on its underlying mechanisms, especially in pediatric populations, is relatively limited (see *Hirsh et al., 2015*; *Hollingshead et al., 2016*; *Miller et al., 2020* for a few exceptions). Drawing upon neuropsychology, social psychology, and pain research, we argue that observers' attentional processing of another's pain may constitute an important mechanism in understanding racialized inequities in pediatric pain care (*Azevedo et al., 2013*; *Dickter & Bartholow, 2007*; *Hoffman et al., 2016*; *Ito, Thompson & Cacioppo, 2004*; *Vervoort & Trost, 2017*). Attention to another's pain is essential for accurately detecting and assessing pain, and appropriate caregiving behaviors (*Vervoort & Trost, 2017*). Indeed, findings of *Vervoort et al. (2013, 2014)* indicate that observers (*i.e.*, parents) more accurately detect pain in the other (*i.e.*, their child) and engage in more protective behaviors when they demonstrate enhanced attention to the other's (*i.e.*, their child's) pain. In the context of racialized biases in pain, our previous work (*Kissi et al., 2022*) showed that White observers attend differently to Black *vs.* White *adults'* facial expressions of pain. Specifically, and counter to expectations, our findings indicated that observers' attention was more easily drawn to Black pain faces. However, the more observers endorsed the false pain belief that White individuals feel pain more easily than Black individuals, the quicker they disengaged from Black *vs.* White pain faces. While intriguing, these findings are preliminary and no study has examined whether observers' attentional processing of White *vs.* Black pain faces manifests (1) similarly or (2) differently in the context of pediatric pain; two possibilities that have merit considering the broader racialized disparities research suggesting that adults are less racially biased towards young children (*Goff et al., 2014*; *Priest et al., 2018*)

and Black children are often victims of adultification (*i.e.*, they are more likely to be treated as adults) (*Baetzel et al., 2019*; *Koch & Kozhumam, 2022*).

The current study had two aims. First, we examined White observers' attentional processing of varying levels of facial expressions of pain demonstrated by White *vs.* Black *children* and observers' estimations of these pain expressiveness levels. Attentional processing was indexed by Attentional Engagement (attention toward pain faces) and Attentional Disengagement (attention away from pain faces) using a Visual Search Task (VST; see *Kissi et al., 2022*; *Notebaert et al., 2011*). Drawing upon prior work indicating that White people have a lower threshold for detecting pain in White people relative to racialized groups (*Mende-Siedlecki et al., 2019*; *Haas et al., 2024*) and research on ingroup favoritism (*Kawakami et al., 2014*; *Zebrowitz, Matthew Bronstad & Lee, 2007*), we expected increased attentional engagement with, reduced attentional disengagement from, and higher pain estimations for White *vs.* Black children's pain faces. Second, we explored whether observers' pain beliefs and children's facial pain expressiveness levels moderated observers' attentional processing and pain estimations.

## MATERIALS AND METHODS

### Ethical approval

All study procedures were approved by the Ethics Committee of the Faculty of Psychology and Educational Sciences of Ghent University.

### Data storage

All data and analytic scripts used for the analyses can be found on Open Science Framework (OSF; see https://osf.io/ydpgs/?view_only=9e26c5c9fb3b47d293621a70ed1019d1).

### Procedure

Participants were recruited *via* Prolific (https://www.prolific.com/), an online recruitment platform. The study inclusion criteria were: (1) having English as a first language, (2) identifying as White, (3) living with a biological child, and (4) having access to an internet-enabled device. To take part in this study, participants were directed to Limesurvey (https://www.limesurvey.org/; an online survey application) where they were first asked to provide their written informed consent, followed by answering questions assessing their demographics and pain beliefs. After answering these questions, participants completed the VST and offered pain ratings *via* Millisecond Inquisit Web (https://www.millisecond.com/products/web/academic), an online tool for conducting research. After completion of the study, participants were debriefed.

### Participants

A total of 99 participants took part in the study. Nineteen participants were omitted from analyses because of an error rate higher than 20% for the visual search task (VST) (*N* = 16) or because their VST or demographic data were missing (*N* = 3) (*Kissi et al., 2022*). The final sample consisted of 80 participants. All participants identified as White[2] and were English speaking. The country or region of residence of most participants was the United

[2] Rooted in imperialistic and colonial times, racism persistently undermines and hinders the health and well-being of racialized individuals worldwide while conferring privilege upon White individuals (*Morais et al., 2022*). With this understanding and our objective to uncover potential mechanisms of racism in pediatric pain care, we chose to investigate the determinants of racism within the group wielding power, namely White observers.

Kingdom (67.50%; 21.30% United Arab Emirates, 6.30% U.S.A., 2.50% Ireland, 1.30% Aruba, 1.30% Korea). Similarly, the majority of the respondents were born in the United Kingdom (67.50%; 20.00% in the United Arab Emirates, 3.80% in Ireland, 1.30% in Jamaica, 1.30% in Aruba, and 6.30% in the United States). The mean age was 43.59 years ($SD$ = 10.15). Forty-two identified as women and 38 as men. Most participants were married or cohabiting (81.30%). The majority had completed higher education (*i.e.*, beyond the age of 18 years; 67.60%), with 32.50% holding a bachelor's degree and 20.00% having completed a master's degree. Additionally, 3.80% held a Ph.D., 2.50% had other advanced degrees, and 1.30% indicated that they did not hold any of the listed educational qualifications. The sample displayed a diverse range of occupations, with the majority working in administrative support roles (15.00%). See Appendix S1 for an overview of all occupations. Participants rated their current health status on a 5-point Likert scale, for which 15.00% of participants indicated "excellent," 31.50% "very good," 38.75% "good," 11.25% "average," and 3.75% "bad." Fifty-eight participants reported to have experienced pain in the last six months for, on average, 31.44 days ($SD$ = 50.95; range = 180). At the moment of testing, participants reported a mean pain intensity level of 1.50 ($SD$ = 2.04; range = 8), on a scale ranging from 0 (no pain) to 10 (worst possible pain). Each participant received £5 for participating in this study.

## Materials

### Pain beliefs

Before completing the VST (see below), participants were asked to report their belief about the difference in pain experience between Black and White individuals. The following question was posed: "Indicate who you think experiences the most pain," with response options ranging from "Black people experience pain more easily than White people" (−5) to "White people experience pain more easily than Black people" (+5).

### Stimuli

At the outset of this study, no suitable database existed that contained standardized images of Black and White children displaying painful facial expressions. To circumvent this issue, our goal was to collect images of children that were relatively consistent in size, quality, and orientation that could then be imported into FaceGen, a software used for generating and manipulating 3D models of human faces. Eventually, for privacy and quality reasons, our initial set of children's faces was created using "This Person Does Not Exist" (https://thispersondoesnotexist.com/), a website which uses artificial intelligence (specifically, a generative adversarial network) to generate photorealistic images of non-existent people. We randomly generated hundreds of faces and saved those that appeared to be children making neutral expressions, facing the camera straight on, and with nothing obstructing the face (*e.g.*, glasses or hair). Given that few, if any, usable Black child images were generated by this site, all saved images were White-appearing.

The approximately 50 resulting images were subsequently imported into FaceGen Modeller Pro, where they could be manipulated in terms of race and gender. In particular,

we sought to make Black and White versions of each imported face by setting the African slider in FaceGen to +1.5 and the European slider to −1.5 for Black versions, while these values were reversed for the corresponding White versions. We also (a) set the gender slider to 0 for male faces and −1 for female faces, (b) set the age slider as low as possible for all faces, and (c) attempted to manually equate any observed differences in face width and brow height that might have covaried with the race sliders in FaceGen. (In some cases, importing a given face revealed that it would not be suitable for further use, due to issues with obstructions or face orientation).

Once imported, standardized, and selected based on the piloting procedures below (see *Avatars*), we were able to render painful expressions on the resulting faces (referred to from here on as *avatars*). See Appendix S2 for an overview of the avatars that were used.

We note that by default, FaceGen stimuli do not have hair. Only a handful of hair options are available in FaceGen, most of which are relatively unrealistic and none of which would be appropriate for child stimuli. As in prior work deploying FaceGen stimuli (*Freeman et al., 2014*; *Mende-Siedlecki et al., 2021*, *2022*; *Stolier & Freeman, 2016*, *2017*) we chose not to add hair to these faces and further, to vignette them in order to focus solely on the face region and minimize the appearance of "baldness."

**Avatars.** In total, 70 images of child avatars were created (35 selected AI generated images (17 boys and 18 girls), with Black and White versions of each one generated in FaceGen). A preliminary study was carried out to validate and select the subset of avatars to be used in the main study. Participants for this preliminary study were recruited from the PSYC100 subject pool at the University of Delaware[3]. These individuals provided informed consent and were compensated with research participation credit. A total of 46 people participated (30 male, 16 female), with a mean age of $M = 18.93$ ($SD = 0.98$). Of them, 17.39% identified as African American, 67.39% as White/Caucasian, 6.52% as Asian, and 8.70% identified with another race.

We sought to ensure that we could select a subset of stimuli wherein (a) the avatars' Black and White versions were robustly perceived as belonging to their intended racial category but did not differ on perceived age or overall racial prototypicality and (b) the boy and girl avatars were perceived distinctly from each other in terms of gender appearance. To collect ratings that would allow us to make these determinations, we adapted a norming procedure primarily from the Chicago Face Database. Each avatar was categorized in terms of its perceived racial identity and sex (*e.g.*, "Indicate the race/ethnicity of this person" (with African American, East Asian, Native American, Pacific Islander, South Asian, White/Caucasian, and Other as multiple choice options; Other was accompanied by a textbox) and "Indicate the gender[4] of this person" (with Female, Male, and Other as multiple choice options; Other was accompanied by a textbox)), as well as rated on age (*e.g.*, "Estimate the approximate age of this person," rated on a scale from 0 to 60) and perceived racial prototypicality (*e.g.*, "How (BLANK) looking are this person's physical features?", rated on a scale from 1 to 5, with the participants' selection from the racial identity categorization question piped into the blank). Ultimately, 12 avatars were selected

[3] Note, that these individuals did not participate in the current study.

[4] We note that while this question was worded so as to ask about perceived gender, the provided options (*i.e.*, Female, Male) referred to biological sex categories.

for use in our primary study (six boys and six girls, 50% White and 50% Black) from the larger set of 35 generated faces. The selected avatars were chosen to be similar in perceived age, generally perceived as their intended race and sex, and roughly equivalent across race in terms of racial prototypicality. These avatars were rated as looking about nine years old on average ($M = 9.07$, $SD = 1.68$), and did not differ significantly in age judgments across race ($M_{Black} = 9.00$, $SD_{Black} = 1.39$; $M_{White} = 9.14$, $SD_{White} = 2.07$; $t(5) = 0.17$, $p = 0.87$) or sex ($M_{Girls} = 9.29$, $SD_{Girls} = 1.41$; $M_{Boys} = 8.84$, $SD_{Boys} = 2.04$; $t(10) = 0.45$, $p = 0.67$). Moreover, the selected boy avatars were more likely to be categorized as male compared to the selected girl avatars (male avatars: $M = 0.82$, $SD = 0.07$, female avatars: $M = 0.49$, $SD = 0.06$; $t(5) = 3.85$, $p = 0.00$), while the selected girl avatars were more likely to be categorized as female compared to the selected boy avatars (male avatars: $M = 0.18$, $SD = 0.07$, female avatars: $M = 0.50$, $SD = 0.19$; $t(10) = -3.71$, $p = 0.00$). In addition, the race/ethnicity categorizations showed that the White avatars were more likely to be perceived as White than Black, $t(10) = 93.94$, $p < 0.001$, and the Black avatars were more likely to be perceived as Black than White, $t(10) = -67.99$, $p < 0.001$. Finally, the selected White avatars were rated as being similarly racially prototypic as the selected Black avatars ($M_{Black} = 4.18$, $SD_{Black} = 0.09$; $M_{White} = 4.15$, $SD_{White} = 0.15$; $t(5) = 0.38$, $p = 0.72$).

**Pain expressiveness levels.** One expression of pain was taken from the empirically validated Delaware Pain Database (DPD; *Mende-Siedlecki et al., 2020*) and selected to be rendered on all 12 avatars chosen based on the preliminary study. This expression was created in FaceGen Modeller Pro by manipulating a subset of 100 sliders corresponding to action unit movements (*e.g.*, brow lowering, eyelid tightening), larger scale expressions (*e.g.*, anger, disgust), and mouth movements associated with specific phonemes (*e.g.*, ooh, aah). In this case, the selected expression was composed of adjustments to sliders representing sadness, closed smile, sneering, squinting, and the production of two phonemes ("B, M, P" and "F, V"). See Appendix S2 for the slider values.

This expression was previously normed in terms of its resemblance to pain and a series of other emotional expressions (full details are available in the DPD Supplemental Materials; see Study 2 therein, "Additional information regarding stimulus norming"). This expression was rated above the scale midpoint (1 to 7 scale) in terms of painfulness ($M = 5.59$, $SD = 1.68$), and was rated as resembling pain more than any other emotion (average other emotion $M = 2.23$ (the highest rating received was for sadness; $M = 2.87$, $SD = 1.76$); all $p$s from $t$-tests comparing pain *vs* each other emotion $< 1E–08$).

For each individual avatar, three levels of facial expression of pain were created: no pain/neutral, moderate pain, and high pain. The neutral level expression was simply the baseline avatar's face with no expression rendered on it. The high pain level expression was created by rendering the selected pain expression at full intensity on the avatar. Finally, the moderate pain level expression was designed to be two-thirds as intense as the high pain expression. To create this, we multiplied each final slider value in the high pain expression by 0.67. For example, the "sneer" slider was set to 1 in the high intensity version of the expression and 0.67 in the moderate pain level version.

## Visual search task

To examine potential differences in observers' attention to pain shown by Black *vs.* White children in pain, participants were asked to perform a VST (see *Kissi et al., 2022*). This design was chosen because it corresponds to real-life settings in which multiple stimuli compete for attention, and not just pain *vs.* no pain stimuli. The VST was programmed using Inquisit 5.0 and comprised 12 practice trials and two blocks of 144 test trials. At the beginning of each trial, a black fixation cross was presented for a duration of 1,000 ms at the center of a white screen. This fixation cross was replaced by a set of images each presented in a circle (image size: 2.5 cm × 2.5 cm) of either Black or White faces with one of the three pain expressiveness levels, with a distractor (tilted right or left line) or target stimulus (horizontal or vertical line) shown on the child's forehead.

On each trial, participants were asked to press a key as quickly and accurately as possible indicating whether the target stimulus comprised a horizontal line (pressing key "4") or vertical line (pressing key "6"). There were three different trial types: congruent, incongruent, and neutral trials. All trial types consisted of six child faces (*i.e.*, avatars), of which the specific composition differed between the types. Images of only White faces were shown during half of the practice and test trials; the other half of the trials comprised only Black faces.

Congruent trials consisted of the target stimulus shown on the forehead of a child's face expressing either moderate or high pain. The remaining five faces presented in each trial were neutral faces paired with distractor stimuli. Incongruent trials consisted of the target stimulus shown on the forehead of a child displaying a neutral facial expression, whereas the other five child facial images (comprised of one painful facial expression and four neutral facial expressions) were combined with a distractor stimulus. Only one of the six faces on each congruent and incongruent trial represented a pain face. Neutral trials consisted of six neutral faces, of which five were combined with a distractor stimulus and one was combined with a target stimulus. See Fig. 1 for an example of the three trial types.

To ascertain participants could not strategically use the pain faces to localize the target stimulus, the 1/n procedure was used whereby the number of congruent trials within each block was restricted to 1 divided by the number of possible locations (*i.e.*, 6 in the current study) where a pain-related stimulus could be presented (*Kissi et al., 2022*). Accordingly, each of the two blocks of the VST consisted of 12 congruent, 60 incongruent, and 72 neutral trials (with a max. trial duration = 5,000 ms, error feedback = 500 ms, and inter-trial duration = 500 ms). To contextualize the pain faces for this study, the following information was presented to the participants on screen:

"*In a few seconds you will see a few photographs of children's faces. Some of these photographs will consist of children who expressed pain during common pain procedures (e.g., vaccinations, wound care, and blood drawing procedures)*".

## Pain ratings

After completing the VST, participants estimated the pain expressiveness level of each avatar (image size 6.5 × 6.5 cm) on a numerical rating scale. This scale ranged from 0

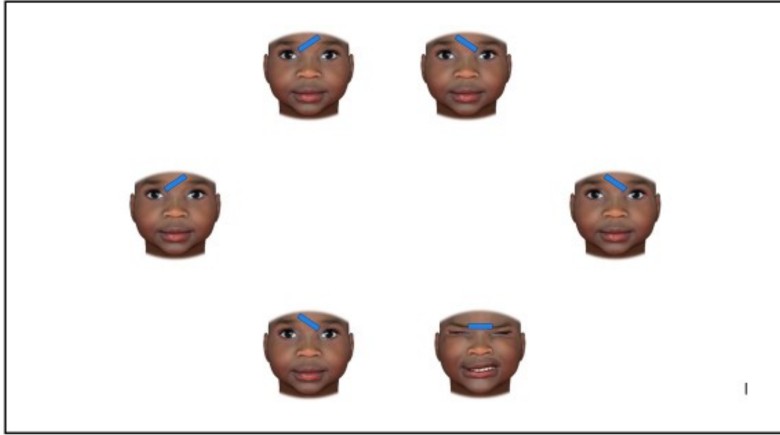

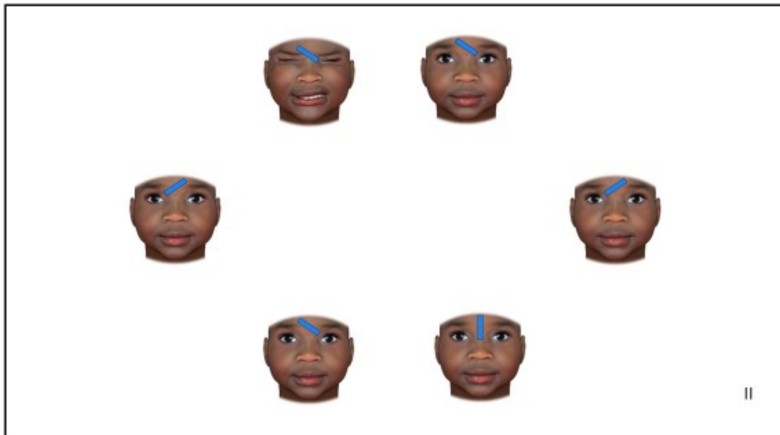

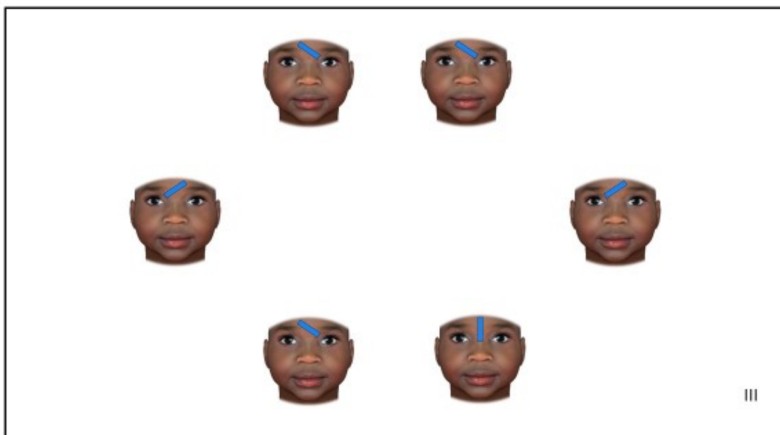

**Figure 1 Examples of the three trial types (I. congruent trials, II. incongruent trials, and III. neutral trials) where facial expressions of a Black girl were presented.**

(indicating no pain) to 10 (indicating a lot of pain). The avatars were presented in random order.

## Data reduction and preparation

VST trials with incorrect responses (3.57%) and outliers (1.43%) were removed before data analyses. Outliers were defined as trials for which the reaction time (RT) was 2.5 standard deviations slower or faster than the mean RT for that particular participant. Pain belief values and age were standardized across persons and gender was contrast coded (woman = −0.50, man = 0.50). In addition to a dummy variable for race (White = 1.00, Black = 0.00), a contrast coded variable for race (White = 0.50, Black = −0.50) was created to include a more interpretable random slope for race.

## Statistical plan

To compare the attentional bias to pain when viewing Black *vs* White children's faces, the VST reaction times were analyzed within a (generalized) linear mixed model ((G)LMM) framework. The statistical plan consisted of three steps.

First, five different modeling approaches (three LMM and two GLMM) were compared to select the most appropriate model for the reaction times. The first LMM modelled raw reaction times, the second LMM modelled the natural logarithm of the reaction times, and the third LMM modelled the "optimal" Box-Cox transformed reaction times. The lambda parameter of this "optimal" Box-Cox transformation was the optimal lambda obtained from using a linear regression model (which corresponds with the LMM with only the fixed effects and no random effects). The first of the two GLMM was a Gaussian GLMM for the raw response times with a log link, the second an inverse Gaussian GLMM with an identity link. All models included fixed effects for *gender*, *age*, and *pain belief* at the between person level, and for *race* (White *vs*. Black), *trial condition* (a combination of trial type and pain expressiveness: congruent—high pain; congruent—moderate pain; neutral; incongruent—high pain; incongruent—moderate pain), and their interaction at the within person level. In addition, within-between two- and three-way interactions including *pain belief*, *race* and *trial condition* were included. The random effect structure included a random intercept and random slopes for *race* and *trial condition*. To select the best model, diagnostic plots were compared between the modeling approaches.

Second, using the most appropriate model from step one, the need for the two- and three-way interactions including *pain belief* and *trial condition* on the response times was investigated by comparing the model with and without these interactions using a chi-square likelihood ratio test.

Finally, using the best fitting model (*i.e.*, with or without the interactions), contrasts were used to test the following specific hypotheses: (a) there is attentional engagement with pain (regardless of race), (b) there is attentional disengagement from pain (regardless of race), (c) attentional engagement with pain differs for Black and White faces, and (d) attentional disengagement from pain differs for Black and White faces. Particularly, attentional engagement was operationalized as the difference between the expected reaction times on neutral *vs*. congruent trials. Attentional disengagement was

operationalized as the difference between the expected reaction times on neutral *vs.* incongruent trials (see Appendices S4–S5 for contrast weights per trial condition). In addition, the possible impact of the level of pain expressiveness on attentional engagement and disengagement differences between Black and White faces was investigated using additional contrasts (see Appendix S6 for the used contrast values in the final steps of the analyses). These analyses were executed in R 4.5.0 (*R Core Team, 2025*) using the lme4 and lmerTest packages (for fitting the models) and the emmeans package (for the contrast tests). As requested by the editor, we refrained from providing *p*-values for the LMM results, because the sampling distributions of the estimates under the null-hypotheses are unknown for finite sample sizes (*Epifania, Anselmi & Robusto, 2024*; *Bates et al., 2015*).

Furthermore, the effect of race (Black *vs.* White) and pain expressiveness level (no pain, moderate pain, high pain) on pain estimations as dependent variable was examined using a $2 \times 3$ repeated measures analysis of variance (RM ANOVA). The moderating effect of pain beliefs on pain estimations was examined by adding pain beliefs as a covariate to the RM ANOVA. Skewness and Kurtosis values of dependent variables were in most instances close to zero and in all instances between −2 and 2 suggesting normality (*Hair et al., 2022*). We did not control for the effects of participants' demographics (*i.e.*, age, gender) as these variables did not impact participants' pain ratings. These analyses were performed using SPSS version 29. Alpha was set at 0.05 for all statistical tests and either the partial eta-squared or Cohen's *d* was used to report effect size. Greenhouse-Geisser corrections were performed in case the sphericity assumption was violated (Mauchly's test of sphericity was $p < 0.05$).

## RESULTS

### Pain beliefs

One-sample t-test indicated that participants did not believe that White individuals experience pain more easily than Black individuals or *vice versa* ($M = 0.06$, $SD = 0.83$, $t(79) = 0.67$, $p = 0.25$, Cohen's $d = 0.08$).

### Model selection

Comparison of the five different modeling approaches showed that the Gaussian GLMM with a log link and the inverse Gaussian GLMM with an identity link should not be further considered because of convergence problems. Further comparison of the diagnostic plots showed that the LMM for the natural logarithm of the response times was the most appropriate model approach. All diagnostic plots are available in the Supplemental Materials (Appendices S12–S14).

### VST analyses

Using the LMM for the log response times, the likelihood ratio test indicated that the within-between interactions including *pain belief* did not increase the fit of the model ($\chi^2_9 = 4.55$, $p = 0.88$;[5] see Appendices S9 and S10 for an overview of all parameter estimates for the fixed and random effects, respectively). Therefore, the following tests were performed without the within-between interactions including *pain belief*.

[5] Similar tests for the other two LMM approaches resulted in the same conclusion: no added value of the within-between interactions, and only evidence for a race effect for attentional engagement with and not for disengagement from pain.

Results indicated that there is evidence for between person differences in the VST reaction times. Overall, women had faster reaction times than men ($t(76.50)= -2.85$, $SE = 0.03$), older children were slower ($t(76.60) = 2.27$, $SE = 0.02$), and participants with increased *pain belief* scores tended to have slower overall VST reaction times ($t(76.9) = 2.39$, $SE = 0.02$).

Furthermore, contrast analyses (see also Appendix S7) just failed to support an overall difficulty to disengage from pain-related facial expressions ($z = 1.82$, $p = 0.068$). No evidence was found for an overall increased attentional engagement with pain-related facial expressions ($z = -1.00$, $p = 0.318$).

Results did show evidence for a difference in attentional engagement with Black *vs.* White pain faces ($z = 2.54$, $p = 0.011$), indicating that participants were quicker to engage with pain expressions of Black compared to White avatars (see Fig. 2). As a follow-up, the presence of attentional engagement with White and Black pain faces was tested separately (see Appendices S5 to S8 for the used contrasts weights and test results). Results revealed a significantly faster attentional engagement with Black pain faces only ($z = -2.45$, $p = 0.014$). No difference was found in attentional disengagement from Black *vs.* White pain faces ($z = 0.12$, $p = 0.907$; see Fig. 2).

Exploratory contrast analyses further indicated that the difference in attentional engagement with Black *vs* White pain faces was particularly apparent for high pain levels ($z = 2.25$, $p = 0.024$), but not for moderate pain levels ($z = 1.48$, $p = 0.140$; see S11 for the results of the performed contrast tests).

## Pain ratings

A $2 \times 3$ RM ANOVA with pain estimations as dependent variable and race (Black *vs.* White) and pain expressiveness level (high pain, moderate pain, no pain) as within subject factors indicated a significant main effect of pain expressiveness level, $F(1.71, 135.19) = 563.27$, $p < 0.001$, $\eta_p^2 = 0.88$, such that pain estimations were highest for faces expressing high pain ($M = 7.44$, $SD = 1.67$) and significantly different from faces expressing moderate pain ($M = 4.82$, $SD = 1.73$), $t(79) = 16.83$, $p < 0.001$, Cohen's $d = 1.89$, and no pain ($M = 1.35$, $SD = 1.36$), $t(79) = 28.23$, $p < 0.001$, Cohen's $d = 3.16$. Pain estimations for moderate pain faces were also significantly higher than estimations of no pain faces, $t(79) = 20.51$, $p < 0.001$, Cohen's $d = 2.29$. Neither a significant main effect for race, $F(1, 79) = 1.91$, $p = 0.17$, $\eta_p^2 = 0.02$ ($M_{Black} = 4.50$, $SD_{Black} = 1.31$; $M_{White} = 4.57$, $SD_{White} = 1.30$), nor a significant interaction effect with pain expressiveness level was found, $F(1.82, 143.84) = 1.38$, $p = 0.25$, $\eta_p^2 = 0.02$. Adding observers' pain beliefs as a covariate to the analyses did not reveal any other significant main or interaction effects (all $F \leq 2.15$). See Appendix S3 for the means and standard deviations of the pain ratings as a function of race and pain expressiveness level.

## DISCUSSION

Evidence highlights the existence of racism in pediatric pain care. However, relatively little empirical evidence is available regarding the underlying mechanisms that may account for these inequities (see *Hirsh et al., 2015*; *Hollingshead et al., 2016*; *Miller et al., 2020* for a few

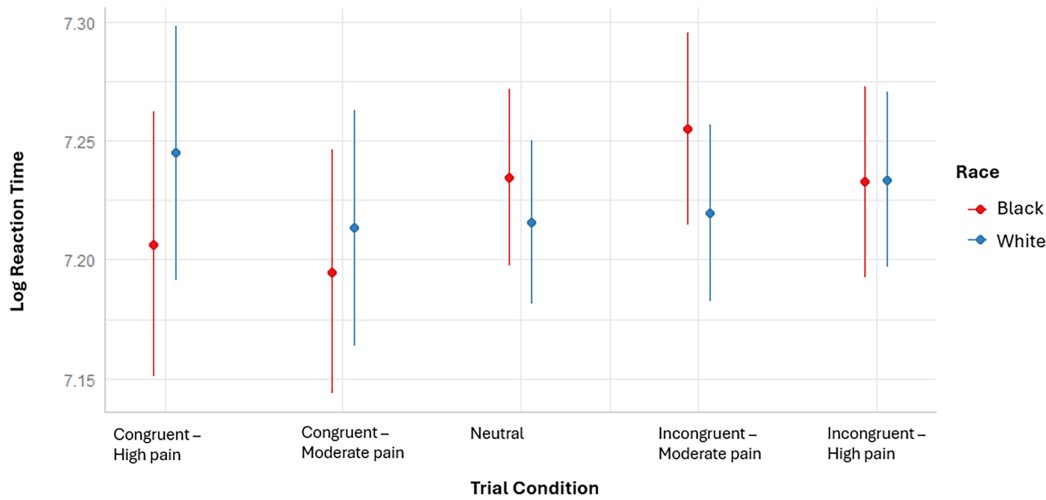

**Figure 2 Representation of the model based mean (error bars) log reaction times of the VST per trial condition and avatar race.**

exceptions). The present study aimed to fill in this gap by examining differences in White observers' attentional processing (*i.e.*, attentional engagement and disengagement) and pain estimations of White *vs.* Black children's facial expressions of pain. Further, we investigated the moderating impact of observers' pain beliefs (*i.e.*, "White people feel pain more easily than Black people" *vs.* "Black people feel pain more easily than White people") and children's facial pain expressiveness levels on observers' attentional processing and pain estimations of White *vs.* Black children. Our findings can be summarized as follows. First, White observers showed faster attentional engagement with Black compared to White pain faces, particularly at high pain intensity levels. Second, no significant difference was found in White observers' attentional disengagement from Black relative to White pain faces. Third, White observers' pain beliefs did not differently impact their attentional processing of White *vs.* Black pain faces. Finally, although White observers' pain estimations increased with increasing pain expressiveness, these estimations did not differ for White *vs.* Black children, nor were they moderated by observers' pain beliefs.

First, we hypothesized that observers' attentional processing of facial expressions of pain would differ for Black and White children. Specifically, we expected that observers' attention would be more easily drawn to White *vs.* Black pain faces (*i.e.*, facilitated attentional engagement) and that they would experience more difficulty diverting their attention from White *vs.* Black pain faces (*i.e.*, impaired attentional disengagement). Interestingly, we observed the opposite effect for attentional engagement with Black *vs.* White faces, particularly when these Black faces expressed high levels of pain, and no differential effect for attentional disengagement from these pain faces. These findings align with our previous findings (*Kissi et al., 2022*) indicating a facilitated attentional engagement towards Black pain faces.

We previously argued that these findings could be explained by observer-driven mechanisms such as threat perceptions (see *Kissi et al., 2022*). Research has shown that

stimuli perceived as threatening capture attention faster than non-threatening stimuli (*Koster et al., 2004*; *Pilch et al., 2020*). This has also been demonstrated in the context of pain (*Eccleston & Crombez, 1999*; *Vervoort et al., 2012*). Further, prior evidence suggests that when observers endorse Black-danger stereotypes, they may perceive pain expressed on Black faces as more threatening compared to pain expressed on White faces (*Donders, Correll & Wittenbrink, 2008*). In light of this, it may be that in the current study, observers' attentional engagement with high levels of pain expressed on Black *vs*. White faces was significantly enhanced because these faces were perceived as more threatening, potentially due to the activation of racialized stereotypes (*e.g.*, "Black people are dangerous").

Another explanation—though not mutually exclusive—is that Black faces expressing high levels of pain elicited differential empathic responses compared to White pain faces expressing the same intensity of pain, with Black pain faces potentially eliciting more concerns about potential threats to *oneself*, rather than to the person in pain. This could be driven by a self-oriented perspective (*i.e.*, imagining how participants themselves would experience the pain) as opposed to an other-oriented perspective (*i.e.*, imagining what it is like for the other person to experience the pain). In particular, when observing others in pain (*i.e.*, a potential signal of threat), neural representations of one's own pain (*Lamm, Decety & Singer, 2011*) are activated and self-oriented aversive emotions are evoked (*Caes et al., 2012*). These emotions appear to be especially enhanced when observers adopt a self-oriented perspective rather than an other-oriented perspective (*Lamm et al., 2008*). Further, recent empirical inquiry suggests that the adoption of a self-oriented *vs*. other-oriented perspective may lead to more initial attention allocation to facial expressions of the person in pain (*Pilch et al., 2020*). However, as we did not examine observers' threat perceptions nor perspective taking, future research is needed to investigate whether and how such an account might explain the current findings.

Future research could also examine whether observers' motivation to respond without prejudice may underlie facilitated attentional engagement towards Black *vs*. White pain faces. Specifically, previous empirical studies have demonstrated that observers motivation to respond without prejudice can influence attentional processing (*Bean et al., 2012*; *Cassidy et al., 2019*). Indeed, *Bean et al. (2012)* found that observers' external motivation to respond without prejudice towards Black people (*i.e.*, stemming from a fear of the negative social consequences of appearing prejudiced towards Black people) was associated with a greater initial attentional bias towards Black *vs*. White faces. In contrast, no significant differences in attentional processing emerged for observers with low external motivation to respond without prejudice towards Black people. It is thus possible that, in the current study, observers demonstrated facilitated attentional engagement towards pain faces of Black *vs*. White children because they were highly motivated to respond in a non-prejudiced manner.

In parallel to the current findings concerning attentional disengagement, our previous findings (*Kissi et al., 2022*) indicated no significant main effect of target race on observers' attentional disengagement. Nevertheless, our prior results revealed a contingent relationship between observers' attentional disengagement from facial expressions of pain and their false pain beliefs. Specifically, observers who endorsed a stronger belief that

White individuals experience pain more easily than Black individuals had less difficulty disengaging their attention from Black *vs.* White pain faces (*Kissi et al., 2022*). However, it is important to note that our previous study focused on adults in pain. Evidence suggests that the relationship between observers' beliefs and their perceptions of pain in adults might not generalize to child targets. For instance, *Miller et al. (2020)* found that while observers' explicit and implicit race-related pain beliefs influenced perceptions of their own pain, these beliefs did not impact their perceptions of children's pain.

One possible explanation for this discrepancy is that, compared to adults, children may elicit different cognitive and emotional responses in adult observers. For instance, adult observers might perceive children as more vulnerable or less likely to exaggerate their pain, prompting more careful pain assessment even if they hold racially biased pain beliefs. Alternatively, social norms emphasizing the need to protect children could override biases typically seen when adult observers perceive pain in other adults (*Haas et al., 2024*; *Kissi et al., 2022*; *Mende-Siedlecki et al., 2019*). While these explanations are tentative, the current findings suggest that, unlike for adult targets (see *Kissi et al., 2022*), observers' pain beliefs may not be related to their attentional processing of children's facial expressions of pain. Future research is needed to investigate the mechanisms underlying this potential age-related difference.

Contrary to prior research indicating racialized disparities in observers' pain estimations among adult samples (*Anderson, Green & Payne, 2009*; *Kissi et al., 2022*), our study did not find evidence for a significant main effect of the child's race on observers' pain estimations. While this finding somewhat corroborates the broader racism literature suggesting that biases in stereotype application may be reduced when considering young children (*Goff et al., 2014*; *Priest et al., 2018*), other empirical work indicates that biases associated with racialized stereotypes are observed even towards children as young as 5 years old (*Todd, Thiem & Neel, 2016*). Furthermore, within the context of pediatric pain, *Miller et al. (2020)* demonstrated racialized disparities in observers' assessments of children's pain-related experience. Specifically, they found that Black pediatric pain patients were rated as more distressed and experiencing greater pain-related interference compared to their White counterparts. It is important to highlight that their study did not specifically examine racialized disparities in observers' pain intensity estimations based on the children's facial expressions, which was the focus of our current study and is of considerable clinical importance (*Palermo et al., 2021*). To our knowledge, our study is the first to directly compare observers' estimations of pain intensity based on facial expressions demonstrated by Black and White children. This limits our ability to make direct comparisons with previous findings and underscores the need for further research in this area.

A number of study limitations should be noted. First, attention was measured indirectly and statically, using reaction times. Eye-tracking methodology may constitute an alternative method to more precisely assess observer attention for pain expressions as it allows to not only investigate attentional processing more *directly* (rather than statically and indirectly *via* response times) but also more *dynamically* (*i.e.*, continuously over time) (*Chan et al., 2020*; *Vervoort et al., 2013*). Examining the time course of observer attention

to child pain allows more precise insight into initial *vs.* later attentional deployment. Prior evidence indicating vigilant-avoidant attentional patterns towards Black but not White faces (*Bean et al., 2012*) suggests this might be particularly relevant for understanding racialized inequities in pain care. Accordingly, future research employing more direct and dynamic assessments of observer pain-related attention, such as eye-tracking, is warranted.

Second, participants were drawn from the general population, which may limit the practical applications of the findings. Given the underrepresentation of healthcare professionals in our sample (see Appendix S1), future research specifically focusing on this population is warranted to better understand attentional processing of pain among healthcare providers. This research could additionally examine potential differences in White *vs.* racialized health care providers' attentional processing to further our understanding of racism in pain care. Relatedly, future research is needed to examine how observed differential attentional patterns as well as proposed associated mechanisms translate into actual caregiving behavior accounting for racialized disparities in pain care. Examining, for example, how observer perspective taking (self *vs.* other) modulates racialized disparities in pain care *via* observer attention towards the sufferer's pain is one potential avenue that might provide invaluable insights and recommendations for empathy-enhancing interventions (see *e.g.*, *Hirsh et al., 2019*; *Wandner et al., 2015*) aimed at promoting equitable and inclusive pain care.

Third, we used avatars making only one specific expression of pain, instead of a variety of images of genuine child pain expressions. While avatars allow for standardization of facial features and are reliable and valid stimuli to investigate inequities in pain care (*Hirsh et al., 2019*; *Kissi et al., 2022*; *Miller et al., 2020*; *Wandner et al., 2012*), they may not fully capture the richness and complexity of real human facial expressions (*Wilson & Soranzo, 2015*) that may be critical to observers' perceptions. Real-life pain expressions often involve dynamic facial movements that occur in specific contexts. These dynamic and contextual elements add depth and meaning to the expressions, that may influence how observers process and asses pain. The absence of these elements may, therefore, have important implications for the applicability of our findings to real-life settings. Indeed, while previous studies have demonstrated that observers (*i.e.*, health care providers) have reported that they respond to and treat pain-expressing avatars similar to how they deal with real patients (*Wandner et al., 2015*) and that avatars can be used for research on complex social interactions (see *e.g.*, *Johnsen et al., 2006*), it should be noted that they are not perfect substitutes for naturally occurring interactions. Furthermore, while certain facial action units are typically associated with expressions of pain, there is no one canonical pain expression–there is considerable variability across individuals, groups, situations, and pain contexts (*Kunz, Meixner & Lautenbacher, 2019*). Future research is needed to examine whether our findings generalize to genuine and varying configurations and intensities of child facial pain expressions as well as to explore the pattern of results that may emerge from using real-life pain expressions *vs.* those generated through artificial intelligence.

Fourth, we assessed observers' beliefs regarding Black *vs.* White *individuals* rather than *children* in pain, which may not be predictive of observers' attentional processing of children's pain. Future research should therefore assess how racialized pain beliefs

about children relate to observers' attentional responses to pain expressed by Black *vs*. White children.

Finally, although our experimental design aimed to minimize the impact of confounding variables through standardization and randomization, unmeasured factors may still have influenced our results. One such factor that may be particularly relevant is implicit racial bias as contemporary racism is often expressed implicitly (*Pearson, Dovidio & Gaertner, 2009*) and prior work has shown that pediatricians' implicit biases are associated with pain treatment recommendations for youth (*Sabin & Greenwald, 2012*). As such, we recommend that future research with larger and more diverse samples examine the role of implicit racial biases and other potentially relevant factors in shaping racialized disparities in observers' attentional processing and pain estimations.

## CONCLUSIONS

This study represents the first investigation into observers' attentional processing of facial expressions of pain demonstrated by Black *vs*. White children. Preliminary evidence was found for racialized disparities in observers' attentional processing of pain, highlighting its potential significance in understanding racism in pediatric pain care. To move the field forward, future studies using healthcare providers are needed to examine their attentional processing and the extent to which this processing predicts actual pain care decisions. Such investigations are pivotal to advance our understanding of racism in pediatric pain care and inform interventions that promote more equitable and inclusive care.

## ACKNOWLEDGEMENTS

The authors would like to thank Jingrun Lin for developing the avatars, generated using artificial intelligence. The random face generator on the website 'This Person Does Not Exist' (https://this-person-does-not-exist.com/en) was used to generate the initial set of faces. This tool employs artificial intelligence—specifically a generative adversarial network—to create photorealistic images of fictitious people.

### Funding
This work was supported by a FWO Grant (Research Foundation Flanders, n° G023423N) awarded to Tine Vervoort, Dimitri Van Ryckeghem & Adam Hirsh. The funders had no role in study design, data collection and analysis, decision to publish, or preparation of the manuscript.

### Grant Disclosures
The following grant information was disclosed by the authors:
FWO Grant (Research Foundation Flanders): n° G023423N.

### Competing Interests
The authors declare that they have no competing interests.

## Author Contributions

- Ama Kissi conceived and designed the experiments, performed the experiments, analyzed the data, prepared figures and/or tables, authored or reviewed drafts of the article, and approved the final draft.
- Dimitri Van Ryckeghem conceived and designed the experiments, analyzed the data, authored or reviewed drafts of the article, and approved the final draft.
- Peter Mende-Siedlecki conceived and designed the experiments, authored or reviewed drafts of the article, and approved the final draft.
- Adam Hirsh conceived and designed the experiments, authored or reviewed drafts of the article, and approved the final draft.
- Ischa Van Alboom analyzed the data, authored or reviewed drafts of the article, and approved the final draft.
- Dries Debeer analyzed the data, authored or reviewed drafts of the article, and approved the final draft.
- Tine Vervoort conceived and designed the experiments, authored or reviewed drafts of the article, and approved the final draft.

## Human Ethics

The following information was supplied relating to ethical approvals (*i.e.*, approving body and any reference numbers):

The Ghent University Ethical Board Faculty of Psychology & Educational Sciences approved the study.

## Data Availability

The raw data is available in the Supplemental Files.

## Supplemental Information

Supplemental information for this article can be found online at http://dx.doi.org/10.7717/peerj.19969#supplemental-information.

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
