# Peer review of "Racialized bias in pediatric pain: the role of observers' attentional processing and estimations of children's pain"

_PeerJ, doi:10.7717/peerj.19969_

## Round 0.1 · original submission · Major Revisions

I do really appreciate this contribution for I believe it addresses an extremely urgent topic, that is how potential racial biases affect the perception of pain in pediatric care.

I commend the authors for the incredible work that they have done for generating the stimuli and making them available.

However, I am not convinced about the statistical analyses that they have carried out. Indeed, dealing with response times is not easy given their skewed distributions, and the application of t-tests and anova (which are linear models) is tricky given that the assumptions on which they rely are almost always violated.

I would suggest to re analyze the data with other approaches. For instance, the authors could use a generalized linear mixed effects models (so they can control for the random variability of the respondents and the stimuli) with a gamma distribution and an inverse or a log link function. Another option would be to log-transform the response times and apply a linear mixed effects models on the log-transformed response times.
I ask the authors to perform such analyses within a mixed effects model framework.

Reviewer 1 ·

Basic reporting

This study investigates racial disparities in attentional processing and pain estimations of facial expressions of pain in Black and White children, aiming to shed light on mechanisms underlying racism in pediatric pain care. Through a series of experiments, the researchers examine how White observers engage and disengage their attention when presented with facial expressions of pain from Black and White children.
Contrary to initial hypotheses, the study finds that White observers demonstrate facilitated attentional engagement towards Black pain faces, potentially driven by perceptions of threat or self-oriented perspective.

The introduction demonstrates a strong grasp of the relevant literature, citing numerous studies to support the assertion of racial disparities in pain care across different age groups. This comprehensive review adds to the credibility of the manuscript and provides a good foundation for the research objectives. The authors clearly outline the aims of the study, which focus on investigating White observers' attentional processing of facial expressions of pain exhibited by White and Black children, as well as the potential moderating effects of observers' pain beliefs and children's pain expressiveness levels. The objectives are clearly stated.

Experimental design

The Methods section provides a detailed and transparent account of the study design, participant recruitment, stimuli selection, and data collection procedures. Overall, the methods are well-structured. Here are some specific comments:
The use of an online recruitment platform is appropriate for reaching a diverse sample. The inclusion criteria are clearly defined
The detailed description of the process for selecting and validating stimuli, including avatars and pain expressions, demonstrates rigor in ensuring the quality and appropriateness of the visual stimuli used in the study.
Some suggestions from me:
Participant Demographics: While the sample size is adequate and the demographics of the participants are provided, it would be beneficial to include more information regarding the representativeness of the sample, particularly in terms of socioeconomic status and geographic location.
Similarly, while the statistical analyses are described in detail, it would be helpful to provide more information on potential covariates considered in the analysis, such as participants' demographic characteristics or other relevant factors.

Validity of the findings

Presentation of results is clear, and the authors use appropriate statistical tests and effect sizes.
I would also suggest adding these to discussion:
-While discussing the opposing patterns of attentional engagement to Black and White pain faces, additional clarification on the underlying mechanisms driving these effects would be better
-While the discussion briefly mentions some methodological limitations, such as the indirect measurement of attention using reaction times, please further elaborate on the implications of these limitations. How these limitations may have influenced the findings?
-They point to potential implications of the findings for pediatric pain care, they can also offer some recommendations for promoting equitable and inclusive care based on the study's findings

Additional comments

Overall the manuscript holds promise for making a valuable contribution to the understanding of racialized disparities in pediatric pain care.

·

Basic reporting

The manuscript is written in clear and professional English throughout. The article structure follows academic standards, with an introduction providing sufficient background and context, methods, results and conclusions sections. The authors have appropriately cited relevant prior literature. Raw data have been made available.

The only suggestion I have for improvement in this section is to provide additional details on the stimuli used in the study, specifically the computer-generated child avatars. More information on how these were created and validated would strengthen the methods. But overall, the basic reporting meets the required standards.

Experimental design

The research addresses an original question that is highly relevant and meaningful - examining potential attentional mechanisms underlying racialized biases in observers' perceptions of children's pain. The research question and hypotheses are clearly articulated. The study design and methods, including the visual search task and pain belief measures, are rigorous and described in sufficient detail to enable replication.

The research has been conducted to a high ethical standard, with IRB approval obtained. No major issues noted with the experimental design. The study is well-suited for publication in PeerJ.

Validity of the findings

The authors have made all data available for review. The data analysis and results appear robust and statistically sound. Effect sizes are reported. The conclusions are stated clearly and are appropriately linked back to and supported by the study results. No claims are made beyond what the data shows.

The research provides novel insights into attentional processing of pain expressions in White vs. Black children, an understudied area. Meaningful replication is encouraged given the importance of the topic.

Additional comments

This is a well-conducted and clearly reported study on an important topic with novel findings. A few minor suggestions:

Provide more detail on the computer-generated stimuli in the methods
Discuss limitations of using avatar faces vs. real child faces
Expand a bit more in the discussion on the real-world implications of the observed attentional bias effects for pain care

---

## Round 0.2 · Major Revisions

Although I have explicitly asked the authors to re-run the analysis with models hat are able to deal with both the data structures and the skewed nature of the response times, I see that they did not do that and they did not even ackoweledge my suggestion. I do believe this manuscript has a potential great impact and needs to be published. However, the results MUST be obtained with statistical models that are appropriate for the data under investigation. Please consider asking for help on the statistical analyses.

Furthermore, I kindly ask you to address ALL the issues raised by Reviewer 2.

Reviewer 1 ·

Basic reporting

This study investigates racial disparities in attentional processing and pain estimations of facial expressions of pain in Black and White children, aiming to shed light on mechanisms underlying racism in pediatric pain care. Through a series of experiments, the researchers examine how White observers engage and disengage their attention when presented with facial expressions of pain from Black and White children.
Contrary to initial hypotheses, the study finds that White observers demonstrate facilitated attentional engagement towards Black pain faces, potentially driven by perceptions of threat or self-oriented perspective.

The introduction demonstrates a strong grasp of the relevant literature, citing numerous studies to support the assertion of racial disparities in pain care across different age groups. This comprehensive review adds to the credibility of the manuscript and provides a good foundation for the research objectives. The authors clearly outline the aims of the study, which focus on investigating White observers' attentional processing of facial expressions of pain exhibited by White and Black children, as well as the potential moderating effects of observers' pain beliefs and children's pain expressiveness levels. The objectives are clearly stated.

Experimental design

The Methods section provides a detailed and transparent account of the study design, participant recruitment, stimuli selection, and data collection procedures. Overall, the methods are well-structured.
The use of an online recruitment platform is appropriate for reaching a diverse sample. The inclusion criteria are clearly defined

Validity of the findings

Presentation of results is clear, and the authors use appropriate statistical tests and effect sizes.

Additional comments

Overall the manuscript holds promise for making a valuable contribution to the understanding of racialized disparities in pediatric pain care.

·

Basic reporting

The manuscript is written clearly in professional English, with a logical structure and sufficient references.
The research question is well-defined, and the literature review provides strong context, especially regarding racial bias and pediatric pain perception.

Experimental design

The study design is rigorous and fits well within the journal’s scope. The research question is both novel and relevant.
The methods are described in enough detail to allow replication.But more detail is needed regarding how potential confounding factors (e.g., implicit racial biases) were controlled.

Validity of the findings

The findings are presented without overemphasis on their potential impact, adhering to PeerJ's guidelines. The study's novelty lies in its focus on attentional processing as a potential mechanism for racial bias in pediatric pain assessment, which is a relatively unexplored area in the literature.

Additional comments

The study is a valuable contribution to the field of pediatric pain and racial bias research. Still, there are a few areas where the manuscript could be strengthened. First, the limitations of using AI-generated avatars should be discussed more thoroughly, particularly in terms of how they might differ from real-life pain expressions. Second, while the authors mention that pain beliefs did not significantly affect attentional processing, they could offer more in-depth interpretation and discussion of this finding, as it contrasts with prior research.

---

## Round 0.3 · Minor Revisions

Thank you for performing the requested analysis. I also appreciate that you have addressed the issues raised by R2. However, I still have concerns about the new analyses, since I was not able to find them anywhere, nor are they described in any way. I see they should be in an OSF repository, but I can’t find the link. I found the R code that you have used for performing them, but it still it is not clear what the variable “score” means, because if it is the same dependent variable that you have used in the ANOVA (i.e., the aggregated difference scores, as far as I understand), of course you get the same results – it is indeed the same model. Moreover, while aggregating across trials in each condition might help dealing with the skewness of the data, it should not be done because it does not account for the stimulus variability and the resulting average is inflated by the error variance, this leading to a greater chance of committing type I error (e.g., see Barr et al. 2013). Finally, I would like to see the results of the LMM in the paper, not ones obtained from ANOVAs. While I do understand that people might be more accustomed to reading the results of the ANOVA, this should not be the reason to report the results of data analysis that are not appropriate given the data. So, I strongly recommend to report the results of the LMMs, even if they are not in line with the original results.

Here some literature:
Barr, D. J., Levy, R., Scheepers, C., & Tily, H. J. (2013). Random effects structure for confirmatory hypothesis testing: Keep it maximal. Journal of memory and language, 68(3), 255-278.
Epifania, O. M., Anselmi, P. & Robusto, E., (under review). A guided tutorial on linear mixed-effects models for the analysis of accuracy and response times in experiments with fully-crossed design. Psychological Methods. Advance online publication. doi: https://doi.org/10.1037/met0000708
Epifania, O. M., Anselmi, P., & Robusto, E. (2022). Filling the gap between implicit associations and behavior: A Linear Mixed-Effects Rasch Analysis of the Implicit Association Test. Methodology, 18 (3), 185-202, doi: https://doi.org/10.5964/meth.7155

---

## Round 0.4 · Minor Revisions

Dear Authors,

Thank you so much for addressing my concerns in such a detailed a precise manner. I just have some minor concerns. Regarding Figure 2, is it still the bar plot? Because barplots should not be used to represent the mean of a distribution, given that they hide the shape of the distribution. I would encourage you to use a different representation, such as boxplots or violin plots. If you want to represent the fixed effects of your models, I strongly suggest using the R package ggeffects (It changed my life for the better).

Other minor comments: Please, report the numerical results according to APA 7. I strongly encourage you not to report the p values associated with the t statistics of the LMMs, since the computation of the degrees of freedom is still debated and the interpretation of the p-values themselves might not be straightforward (see Epifania et al. 2024 for a discussion on the topic).

Thank you again for the work you have done

Epifania, O. M., Anselmi, P. & Robusto, E., (2024). A guided tutorial on linear mixed-effects models for the analysis of accuracy and response times in experiments with fully-crossed design. Psychological Methods. Advance online publication. doi: https://doi.org/10.1037/met0000708

---

## Round 0.5 · Minor Revisions

Thank you for the revised version, I appreciate your effort in addressing all the previous concerns.

I still one little concern. As far as I can see, you removed the signficance from the estimates of the fixed effect obtained with the LMMs, as I have suggested (thank you). However, I think it would be better if you could provide the reason why you did that, otherwise the average reader might not understand why you have reported the significance of the GLMMs and not of the LMMs. One sentence can suffice, I have already sent the literature you might cite for writing down this justification in the previous round.

---

## Round 0.6 · accepted · Accept

All issues and concerns have been addressed, and the manuscript is ready for publication.